# Meal Patterns of Older Adults: Results from the OUTDOOR ACTIVE Study

**DOI:** 10.3390/nu14142784

**Published:** 2022-07-06

**Authors:** Imke Stalling, Birte Marie Albrecht, Linda Foettinger, Carina Recke, Karin Bammann

**Affiliations:** Institute for Public Health and Nursing Research (IPP), University of Bremen, 28359 Bremen, Germany; b.albrecht@uni-bremen.de (B.M.A.); foettili@uni-bremen.de (L.F.); c.recke@uni-bremen.de (C.R.); bammann@uni-bremen.de (K.B.)

**Keywords:** meal patterns, older adults, sociodemographic factors, health

## Abstract

Eating habits have a substantial effect on health, not only because of consumed foods and nutrients, but also because of the regularity of meals. This study investigates meal patterns in older adults. Data from 1198 adults (52.8% female) aged between 65 and 75 years, who resided in Bremen, Germany, were included in this cross-sectional study. Using a self-administered questionnaire, daily meals were assessed and categorised into three meal pattern types: ‘regular eaters’ (eating at least three meals a day), ‘meal skippers’ (skipping one meal a day), and ‘irregular eaters’ (skipping more than one meal a day). Descriptive analyses were carried out, stratified by sex and meal pattern types. Most women and men were regular eaters (51.5% and 51.7%, respectively), 33.8% of women and 33.3% of men were meal skippers, and 14.7% of women and 15.0% of men were irregular eaters. Differences between meal patterns were seen with regard to socioeconomic status; self-rated health; body-mass index; hypertension; consumption of self-prepared meals; and consumption of whole-grain products, green vegetables, meat, and alcohol. The results provide first insights into possible associations between meal patterns and sociodemographic and health factors, and can benefit disease prevention and health promotion in older adults.

## 1. Introduction

Eating habits have a substantial effect on health [1] and with advancing age, the risk of malnutrition increases [2]. Malnutrition is defined as a deficiency or surplus of energy, micronutrients, or protein that leads to adverse health effects [3], such as underweight or overweight and obesity, as well as increased morbidity and mortality [4,5]. This is not only caused by the foods and nutrients that are being consumed, but also by the timing, frequency, and regularity of meals [6]. While having three meals per day is still the leading eating pattern in Western European countries [7,8], investigating different meal patterns is important for research and health promotion. Research has shown that regular eating frequency is associated with lower risks of overweight and obesity [9,10,11,12,13], as well as generally more healthy behaviours in adults [9,14]. On the other hand, skipping meals can have detrimental effects on physical and mental health. It is associated with stress, depression, and suicidal ideation in older adults [6]. Several studies have looked closer into the effects of skipping breakfast [15,16,17,18] and have found associations with higher risks for obesity in adults and older adults [10], regardless of culture or ethnicity [19]. Furthermore, increased risks of type 2 diabetes were observed for adults [20] and older women [21], who skip breakfast.

Consuming meals does not only affect health due to nutritional aspects, but also because of the social aspects of having a meal. Studies have shown that older adults who eat alone tend to skip meals more often and present poor dietary behaviour [22], which can increase the risk of malnutrition [6]. They have also found associations between eating alone and higher risks for obesity as well as cognitive decline [22,23]. Furthermore, sociodemographic factors such as marital status, living situation, and socioeconomic status (SES) are associated with the risk of malnutrition in older adults [24]. Studies have also found correlations between low SES and meal skipping [6,25,26,27], as well as lower quality of consumed meals [27].

Assessing meal patterns has become an important research focus, but knowledge regarding older adults is still scarce. Having an extensive understanding of older adults’ meal patterns can be beneficial for preventing a wide range of non-communicable diseases and health risks, and can help in developing health promotion interventions. Therefore, the aim of this study is to provide further insight into the meal patterns of older adults in Germany and their associated sociodemographic factors and health indicators.

## 2. Material and Methods

### 2.1. Study Design and Sample

The sample of this cross-sectional study comprises participants from the cluster-randomised controlled trial OUTDOOR ACTIVE [28], which is part of the German prevention research network AEQUIPA [29]. Data on health and physical activity (PA) were collected at baseline between June 2018 and July 2019. To assess intrapersonal, interpersonal, and environmental determinants of PA, a self-administered paper–pencil questionnaire was used. Additionally, participants were invited to a short physical examination that included measurements of anthropometry, as well as blood pressure, followed by a modified Senior Fitness Test [30]. Lastly, PA was measured objectively over seven consecutive days using accelerometery. The questionnaire and survey materials are available on request from the corresponding author.

Those eligible for participation were all non-institutionalised 65- to 75-year-olds living in one of the eight randomly chosen sub-districts of Bremen, Germany. Address data were provided by the registry office in Bremen in March 2018. In total, 6853 participants were officially registered in the study region. They received a letter inviting them to participate and were later contacted via phone, if the number could be obtained through one of the available registers. After excluding all institutionalised individuals, 6775 eligible participants remained. Of those, 219 were excluded due to acute health problems, and 69 people died. Furthermore, 55 people were excluded because of language barriers, 155 because of moving out of the study region, 2705 were never reached, and 2195 refused to participate. A total of 151 persons of one sub-district were never contacted, since the sample size had already been exceeded at the end of the survey period. Effectively, 1226 participants took part in at least one of the survey parts (response: 19.5%). Of those, 28 had to be excluded from the analyses because of missing data, which led to a study sample of 1198 participants.

### 2.2. Measures

#### 2.2.1. Meal Patterns and Food Frequency

Meal patterns were assessed using the self-administered paper–pencil questionnaire and a self-developed question that was not validated. Participants were asked how often they normally consume breakfast, lunch, and dinner. We further asked for the consumption frequency of afternoon tea, but did not include this in the analyses. Response categories included ‘daily’, ‘only on weekdays’, ‘only on weekends’, ‘several times a week’, and ‘hardly ever’. Additionally, there was an option to check if precise answers are difficult to give, because of eating irregularly. Participants, who stated that they eat at least three meals daily were categorised as ‘regular eaters’. Participants, who only skip one meal daily were categorised as ‘meal skippers’. Participants who skip more than one meal daily or stated that they ate irregularly were categorised as ‘irregular eaters’.

Participant were also asked how often they prepare their meals themselves. Response categories were ‘hardly ever or never’, ‘often’, and ‘mostly or always’. To obtain an insight into their eating habits, the consumption of different foods was assessed using a self-developed food frequency questionnaire (FFQ). Categories comprised the following: ‘never’, ‘once a month or less’, ‘two to three times a month’, ‘once a week’, ‘several times a week’, and ‘(almost) daily’. Since the question on food frequency was originally developed to assess bone health, it comprised the following food items: hard cheese, soft cheese, milk/buttermilk, yogurt/kefir, eggs, whole-grain products, legumes, cabbage/green vegetables, meat, fatty fish, alcohol, and mineral water. To cover carbohydrates, vegetables, protein, and fats, the variables whole-grain products, cabbage/green vegetables (from here on, green vegetables), and meat (no further specifications) were included in this study. Furthermore, alcohol consumption was included as a potential health-compromising behaviour.

#### 2.2.2. Sociodemographic Information

Age, marital status, whether participants have a partner, and whether they live alone were assessed using the self-administered questionnaire with modified questions from the German Health Interview and Examination Survey (DEGS) [31]. For age, the participants stated their year of birth, which was subtracted from the year they filled out the questionnaire. Marital status was assessed using the response categories ‘single/unwed’, ‘married/registered partnership’, ‘divorced’, and ‘widowed’.

For SES, an additive social class index containing education, income, and occupation was calculated and categorised into quintiles, with the first quintile representing a low SES and the fifth a high SES (for more details, see [32]).

#### 2.2.3. Health Indicators

Self-rated health was assessed using a frequently used valid question from the SF-36 [33], with five response categories ranging from ‘bad’ to ‘excellent’. Participants were further asked to state whether they take medications daily and, if they did, to write down which medications they took using a self-developed question. Subsequently, they were coded according to the Anatomical Therapeutic Chemical (ATC) classification [34]. The number of daily medications, as well as drugs used for high blood pressure (ATC codes beginning with C02, C03, or C07-C09), were included in the analyses.

During the physical examination, height and body weight were measured using a Seca 217 mobile stadiometer (Seca GmbH & Co. KG, Hamburg, Germany) and a Kern MPC 250K100M personal floor scale (Kern & Sohn GmbH, Ballingen, Germany), respectively. Body-mass index (BMI) was calculated by dividing body weight (in kg) by the squared height (in m), and was later classified using the World Health Organization’s cut-offs for underweight, normal weight, overweight, and obesity [35].

Blood pressure was measured twice on the right upper arm using an oscillometric monitor (OMRON 705CPII; Omron Healthcare Co., Ltd., Hoofddorp, The Netherlands). Before each measurement, the participants rested for one minute without talking or any disturbances. Hypertension is defined as systolic blood pressure over 140 mmHg or diastolic blood pressure over 90 mmHg [36]. For the classification of hypertension, we used the second blood pressure measurement. For participants, who did not partake in the physical examination, questionnaire responses, if they were diagnosed with hypertension, were used. To assess chronic diseases, a modified question from the DEGS [31] was used.

### 2.3. Statistical Analyses

Descriptive analyses were carried out for all variables. Absolute and relative frequencies were calculated, except for age and number of daily medications. For these variables, means and standard deviations were calculated. All analyses were conducted separately for women and men, and were conducted using SPSS 22.0 (IBM Corp. Armonk, NY, USA).

## 3. Results

Table 1 shows the descriptive characteristics of the study sample by meal patterns. A total of 52.8% of the participants were female. The majority of women and men were regular eaters, with 51.5% and 51.7%, respectively. A total of 33.8% of women and 33.3% of men were meal skippers, and 14.7% of women and 15.0% of men were irregular eaters. Of the meal skippers, most skipped lunch (women: 59.8%, men: 71.8%), followed by dinner (women: 30.8%, men: 22.3%), and breakfast (women: 9.3%, men: 5.9%). On average, women were 69.7 ± 2.9 years old and men were 69.8 ± 2.9 years old. Women who were regular eaters were the oldest a 70.0 ± 3.0 years. Most of the participants were married (women: 53.8%, men: 74.9%), with female meal skippers and male regular eaters showing the highest percentage and irregular eaters of both sexes showing the lowest. Meal skippers stated most often that they had a partner, with 65.1% among women and 90.4% among men. Irregular eaters showed the highest percentages of participants living alone (women: 56.5%, men: 28.6%). Among women, regular eaters (26.8%) and meal skippers (22.9%) were mostly in the lowest SES quintile, whereas irregular eaters mostly pertained to the 3rd SES quintile (29.0%). Conversely, men showed a different distribution, with regular eaters (29.1%) mostly pertaining to the highest SES quintile, and meal skippers (23.9%), as well as irregular eaters (30.6%), to the 4th.

Table 2 presents descriptive results stratified by sex and meal pattern regarding the frequency of consuming self-prepared meals and select food items. While the majority of women (76.1%) consumed self-prepared meals mostly or always, only 42.9% of men did the same. A total of 21.9% of men stated that they hardly ever or never consumed self-prepared meals. Regardless of sex, irregular eaters consumed self-prepared meals the least (women: 62.4%, men: 35.3%) and had the lowest percentages of eating whole-grain products (women: 86.1%, men: 70.2%), green vegetables (women: 53.3%, men: 48.7%), and among women, meat (25.3%) at least several times a week. Alcohol consumption was the highest among female irregular eaters (42.0%) and male meal skippers (59.0%), while female regular eaters (33.6%) and male irregular eaters (44.1%) had the lowest.

Table 3 contains descriptive results regarding different health indicators by meal pattern. Meal skippers had the best self-rated health (women: 33.2%, men: 36.9% very good or excellent). Regarding the worst self-rated health, female irregular eaters (20.4%) and male meal skippers (14.4%) showed the highest proportions. Across all groups, the majority of participants rated their health as good. Women showed only small differences between meal patterns regarding normal weight and overweight. However, irregular eaters had the highest proportions of obese (18.2%) and underweight participants (2.6%). Normal weight differed only slightly between meal patterns among men, while meal skippers showed the highest percentages of overweight participants (49.1%), and irregular eaters showed the least (42.4%). The latter had the highest proportions of obese men (28.8%). While the prevalence of hypertension was similar between meal patterns among women, male regular eaters (63.7%) had the highest prevalence, and irregular eaters (54.1%) had the lowest. Male regular eaters took antihypertensive medications most often (53.9%) and, on average, female regular eaters (2.6 ± 1.9 daily medications) and irregular eaters of both sexes took the most daily medications (women: 2.6 ± 2.3 daily medications, men: 3.2 ± 2.5 daily medications).

## 4. Discussion

The aim of this study was to provide further insight into the meal patterns of older adults in Germany and their associated sociodemographic factors and health indicators. The results showed differences between meal patterns, with irregular eaters especially standing out. Irregular eaters had the highest proportions of not being married, not having a partner, and living alone. Furthermore, the results indicated that they consumed self-prepared meals, whole-grain products, green vegetables, and meat least often. Regarding alcohol, female irregular eaters consumed it most often compared with the other women, and male irregular eaters least often compared to the rest of the men. Among women, irregular eaters mostly pertained to the 3rd SES quintile, and among men, to the 4th. They rated their health less often as very good or excellent than the other meal pattern types and showed the highest proportions of obese participants.

Our results regarding the distribution of meal patterns (51.5% of women and 51.7% of men ate three meals a day) are in line with previous research, which found the three-meals-a-day pattern to be most common [7,8,37]. In line with our results, Krok-Schoen et al. reported lunch to be skipped most [37], although their assessment only referred to one day. Other studies showed that breakfast is the most-skipped meal among adults of all age groups [38,39]. The differing results could stem from dissimilar age groups, cultural differences, or varying methods of assessing meals (e.g., time of day, participant-identified meals, eating occasions, nutritional components) [8,40,41]. For example, in their study in France, Lafay et al. [39] used an eating diary of seven consecutive days, whereby each meal was assessed in detail with the time, consumed products, and amount, among other things. In our study, however, the consumption frequency of meals was assessed with one question in a questionnaire. Since there is no standard method of measurement and the choice of assessment is dependent on several factors, such as the research question, resources, or participants, these differences are a big issue when comparing research results.

In our study, the majority of regular eaters and meal skippers had a partner, whereas irregular eaters were more often single and lived alone, although this was more noticeable among women. Conversely, Kwak and Kim [6] found meal skipping to be associated with having no partner and living alone, and with meal skippers having a lower SES than non-meal skippers. Our results, however, indicated that female regular eaters and meal skippers had the lowest SES, and male regular eaters pertained to the highest SES quintile. While we distinguished between meal skippers and irregular eaters, Kwak and Kim assessed if participants skipped breakfast and/or dinner over a period of two days [6], which could explain differing findings. In line with our results, Lhuissier et al. [7] also found higher education to be associated with eating regularly among men, and Anderson et al. [42] showed associations between education and generally more healthy behaviours and conscious nutrition among older adults.

In our study, the majority of male regular eaters and meal skippers consumed alcohol several times a week or daily, compared to male irregular eaters, who consumed alcohol once a week or less. For regular eaters, a possible explanation could be the tradition of after-work beers in Germany, especially among men, regardless of employment, which could be a part of eating regularly. Women in our study consumed alcohol less often, but irregular eaters stood out with the highest consumption. Since we did not assess the amount or type of alcohol, our results should be interpreted with caution. Previous studies have found that skipping breakfast [16] or skipping dinner [14] are associated with higher alcohol consumption among adults.

Even though previous research found that meal skipping is associated with lower self-rated health among older adults [43] and could have negative effects on mental health [6], meal skippers in our study had the highest proportions of participants rating their health as very good or excellent, and among men, also of rating their health as less good or bad compared to regular and irregular eaters.

Our results indicated differences in BMI between meal patterns, with irregular eaters having the highest proportions of obese participants. Research on possible associations is scarce, but Huang et al. [15] found higher odds of developing obesity for breakfast skippers in adults. Though skipping meals might not necessarily be the cause for obesity, it is associated with poorer health and can be seen as health-compromising behaviour [15].

While the prevalence of hypertension only showed small differences between meal patterns in our study sample, research found that skipping breakfast [44,45], as well as the timing of meals [46], are associated with higher risks of hypertension. Since we did not assess the time and did not have a large enough study sample for further subgroup analyses, future research should examine the possible associations further.

The study has a few limitations that need to be addressed. Because of the cross-sectional study design, statements regarding causation cannot be made, and longitudinal studies researching possible associations between meal patterns and sociodemographic factors, as well as health indicators, are needed. Furthermore, the questionnaire used in the OUTDOOR ACTIVE study was mainly focused on physical activity, which is why diets and fasting were not assessed. Therefore, the reasons behind meal skipping or irregular eating are unknown. Lastly, we had limited data on food intake (i.e., only select food items in the FFQ) since the question on food frequency was designed for assessing bone health.

One strength of this study is the distinction between meal skippers and irregular eaters, which has not often been investigated in previous studies. Furthermore, research on the characteristics of meal patterns, especially in older adults, is scarce, and our study provided further knowledge on this topic. Our results can help understand older adults’ meal patterns better, and can benefit disease prevention and health promotion.

## 5. Conclusions

The findings of this study showed associations of meal patterns with health behaviour, as well as health indicators, among older adults. Irregular eaters of both genders especially had a poorer diet and poorer health outcomes compared to the subjects of other meal patterns. Additionally, the results indicated differences regarding sociodemographic factors, with irregular eaters being single and living alone more often than other participants. The differentiation between meal skippers and irregular eaters revealed that skipping one meal itself is not associated with unhealthy dietary behaviour or poorer health-related outcomes, but that more irregular meal patterns are. While the study has a few limitations (e.g., cross-sectional design and the questionnaire not being validated) that need to be considered when looking at the results, our findings provide more insight into the meal patterns of older adults and their associated sociodemographic factors and health indicators. Further research to replicate the results in other settings and longitudinal studies disentangling cause and effect are needed.

## Figures and Tables

**Table 1 nutrients-14-02784-t001:** Characteristics of the study sample by meal pattern and sex.

	Women			
	Total (*n* = 633)	Regular Eater (*n* = 326)	Meal Skipper (*n* = 214)	Irregular Eater (*n* = 93)
	Mean (SD)			
Age in years	69.7 (2.9)	70.0 (3.0)	69.4 (2.7)	69.6 (3.1)
	*n* (%)			
Marital status				
Married	337 (53.8)	177 (55.0)	119 (55.9)	41 (45.1)
Divorced	127 (20.3)	64 (19.9)	39 (18.3)	24 (26.4)
Widowed	88 (14.1)	49 (15.2)	21 (9.9)	18 (19.8)
Unwed/single	74 (11.8)	32 (9.9)	34 (16.0)	8 (8.8)
Having a partner	391 (62.9)	204 (63.4)	136 (65.1)	51 (56.0)
Living alone	268 (42.6)	136 (42.0)	80 (37.6)	52 (56.5)
Socioeconomic status				
Lowest quintile	154 (24.4)	87 (26.8)	49 (22.9)	18 (19.4)
2nd quintile	139 (22.0)	74 (22.8)	48 (22.4)	17 (18.3)
3rd quintile	125 (19.8)	57 (17.5)	41 (19.2)	27 (29.0)
4th quintile	118 (18.7)	56 (17.2)	43 (20.1)	19 (20.4)
Highest quintile	96 (15.2)	51 (15.7)	33 (15.4)	12 (12.9)
	**Men**			
	**Total** **(*n* = 565)**	**Regular eater** **(*n* = 292)**	**Meal skipper** **(*n* = 188)**	**Irregular eater** **(*n* = 85)**
	Mean (SD)			
Age in years	69.8 (2.9)	69.9 (2.9)	69.9 (2.8)	69.5 (2.9)
	*n* (%)			
Marital status				
Married	417 (74.9)	223 (77.2)	137 (74.9)	57 (67.1)
Divorced	66 (11.8)	30 (10.4)	25 (13.7)	11 (12.9)
Widowed	26 (4.7)	11 (3.8)	8 (4.4)	7 (8.2)
Unwed/single	48 (8.6)	25 (8.7)	13 (7.1)	10 (11.8)
Having a partner	487 (86.8)	250 (86.2)	169 (90.4)	68 (81.0)
Living alone	123 (21.9)	60 (20.7)	39 (20.9)	24 (28.6)
Socioeconomic status				
Lowest quintile	85 (15.0)	41 (14.0)	26 (13.8)	18 (21.2)
2nd quintile	99 (17.5)	58 (19.9)	31 (16.5)	10 (11.8)
3rd quintile	115 (20.4)	56 (19.2)	42 (22.3)	17 (20.0)
4th quintile	123 (21.8)	52 (17.8)	45 (23.9)	26 (30.6)
Highest quintile	143 (25.3)	85 (29.1)	44 (23.4)	14 (16.5)

SD: Standard deviation.

**Table 2 nutrients-14-02784-t002:** Frequencies of consuming self-prepared meals and select food items.

	Women			
	Total (*n* = 633)	Regular Eater (*n* = 326)	Meal Skipper (*n* = 214)	Irregular Eater (*n* = 93)
	*n* (%)			
Consume self-prepared meals				
Hardly ever or never	18 (2.9)	7 (2.2)	8 (3.7)	3 (3.2)
Often	133 (21.1)	56 (17.3)	45 (21.0)	32 (34.4)
Mostly or always	480 (76.1)	261 (80.6)	161 (75.2)	58 (62.4)
Whole-grain products				
Less than once a week	29(4.7)	15 (4.6)	9 (4.2)	5 (5.4)
Once a week	31 (4.9)	11 (3.4)	12 (5.7)	8 (8.6)
Several times a week or daily	567 (90.4)	296 (92.0)	191 (90.1)	80 (86.1)
Green vegetables				
Less than once a week	97 (15.5)	45 (13.9)	32 (15.1)	20 (21.8)
Once a week	135 (21.5)	66 (20.4)	46 (21.7)	23 (25.0)
Several times a week or daily	395 (63.0)	212 (65.7)	134 (63.2)	49 (53.3)
Meat				
Less than once a week	178 (28.3)	75 (23.1)	58 (27.1)	45 (49.5)
Once a week	196 (31.1)	105 (32.3)	68 (31.8)	23 (25.3)
Several times a week or daily	256 (40.6)	145 (44.7)	90 (41.1)	23 (25.3)
Alcohol				
Less than once a week	292 (46.4)	170 (52.5)	85 (40.0)	37 (39.8)
Once a week	107 (17.0)	45 (13.9)	45 (21.1)	17 (18.3)
Several times a week or daily	231 (36.6)	109 (33.6)	83 (39.0)	39 (42.0)
	**Men**			
	**Total** **(*n* = 565)**	**Regular eater** **(*n* = 292)**	**Meal skipper** **(*n* = 188)**	**Irregular eater** **(*n* = 85)**
	*n* (%)			
Consume self-prepared meals				
Hardly ever or never	123 (21.9)	66 (22.8)	35 (18.7)	22 (25.9)
Often	198 (35.2)	90 (31.0)	75 (40.1)	33 (38.8)
Mostly or always	241 (42.9)	134 (46.2)	77 (41.2)	30 (35.3)
Whole-grain products				
Less than once a week	34 (6.1)	13 (4.5)	12 (6.4)	9 (10.8)
Once a week	50 (9.0)	19 (6.6)	15 (8.1)	16 (19.0)
Several times a week or daily	473 (85.0)	256 (88.9)	158 (85.4)	59 (70.2)
Green vegetables				
Less than once a week	96 (17.3)	42 (14.5)	38 (20.3)	16 (19.5)
Once a week	150 (26.9)	70 (24.3)	54 (28.9)	26 (31.7)
Several times a week or daily	311 (55.8)	176 (61.1)	95 (50.8)	40 (48.7)
Meat				
Less than once a week	86 (15.3)	35 (12.1)	33 (17.5)	18 (21.2)
Once a week	126 (22.4)	55 (19.0)	51 (27.1)	20 (23.5)
Several times a week or daily	351 (62.3)	200 (69.0)	104 (55.3)	47 (55.3)
Alcohol				
Less than once a week	171 (30.5)	89 (30.9)	50 (26.5)	32 (38.1)
Once a week	80 (14.3)	38 (13.1)	27 (14.4)	15 (17.9)
Several times a week or daily	310 (55.3)	162 (56.0)	111 (59.0)	37 (44.1)

**Table 3 nutrients-14-02784-t003:** Health indicators by meal pattern and sex.

	Women			
	Total (*n* = 633)	Regular Eater (*n* = 326)	Meal Skipper (*n* = 214)	Irregular Eater (*n* = 93)
	*n* (%)			
Self-rated health				
Less good or bad	93 (14.9)	50 (15.5)	25 (11.8)	18 (19.8)
Good	358 (57.4)	189 (58.7)	116 (55.0)	53 (58.2)
Very good or excel-lent	173 (27.7)	83 (25.8)	70 (33.2)	20 (22.0)
Body-Mass Index				
Underweight	5 (1.0)	2 (0.7)	1 (0.6)	2 (2.6)
Normal weight	246 (46.9)	128 (47.1)	88 (50.0)	30 (39.0)
Overweight	185 (35.2)	98 (36.0)	56 (31.8)	31 (40.3)
Obesity	89 (17.0)	44 (16.2)	31 (17.6)	14 (18.2)
Blood pressure				
Hypertension	323 (51.0)	171 (52.5)	104 (48.6)	48 (51.6)
Antihypertensive medication	213 (40.5)	113 (41.5)	67 (37.9)	33 (42.9)
	Mean (SD)			
Number of daily medications	2.6 (2.0)	2.6 (1.9)	2.6 (2.0)	2.6 (2.3)
	**Men**			
	**Total** **(*n* = 565)**	**Regular eater** **(*n* = 292)**	**Meal skipper** **(*n* = 188)**	**Irregular eater** **(*n* = 85)**
	*n* (%)			
Self-rated health				
Less good or bad	79 (14.1)	41 (14.3)	27 (14.4)	11 (12.9)
Good	301 (53.8)	155 (54.0)	91 (48.7)	55 (64.7)
Very good or excel-lent	179 (32.0)	91 (31.7)	69 (36.9)	19 (22.4)
Body-Mass Index				
Underweight	0 (0.0)	(0.0)	(0.0)	(0.0)
Normal weight	145 (31.5)	78 (33.5)	48 (29.8)	19 (28.8)
Overweight	221 (48.0)	114 (48.9)	79 (49.1)	28 (42.4)
Obesity	94 (20.4)	41 (17.6)	34 (21.1)	19 (28.8)
Blood pressure				
Hypertension	350 (61.9)	186 (63.7)	118 (62.8)	46 (54.1)
Antihypertensive medication	228 (49.7)	125 (53.9)	72 (44.7)	31 (47.0)
	Mean (SD)			
Number of daily medications	3.0 (2.1)	3.0 (2.0)	2.8 (2.1)	3.2 (2.5)

SD: Standard deviation.

## Data Availability

The data presented in this study are available on request from the corresponding author. The data are not publicly available due to privacy restrictions.

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
