# Peer review of "Meal Patterns of Older Adults: Results from the OUTDOOR ACTIVE Study"

_nutrients, 2022, doi:10.3390/nu14142784_

Round 1

Reviewer 1 Report

Overview

The manuscript titled Meal patterns of older adults: results from the OUTDOOR ACTIVE study describes the insightful examination of meal patterns of older adults. The study has been completed with necessary data. However, the author requires to complete the following queries before publication.

Comments

In abstract, Line no. 9. The sentence should be rewrite. I.e. Data of 1198 65-75…..study.

All keywords should start with uppercase

The introduction part should be improved by including some more information.

The author should check typos in the entire manuscript. For example. The percentage symbol should be joined to the respective value.

The conclusion part should be improved 

Author Response

Overview

The manuscript titled Meal patterns of older adults: results from the OUTDOOR ACTIVE study describes the insightful examination of meal patterns of older adults. The study has been completed with necessary data. However, the author requires to complete the following queries before publication.

Response: Thank you very much for your comments and suggestions. We have adapted the manuscript accordingly and hope it has improved.

Comments

In abstract, Line no. 9. The sentence should be rewrite. I.e. Data of 1198 65-75…..study.

Response: We have rewritten this sentence, so it should be easier to read.

All keywords should start with uppercase

Response: We have decided to write them all in lower case, since the already published papers of this special issue all have exclusively lower case keywords.

The introduction part should be improved by including some more information.

Response: We have added further information to the introduction.

The author should check typos in the entire manuscript. For example. The percentage symbol should be joined to the respective value.

Response: We have removed the spaces between values and percentage symbols. We have further checked the entire manuscript for typos and have corrected them accordingly.

The conclusion part should be improved 

Response: We have completely rewritten the conclusion for a better summary of the results and therefore a better understanding.

Reviewer 2 Report

This is a cross-sectional study which is associated with the meal patterns of older adults. Researchers have categorized older adults into three meal patterns types (regular eaters, meal skippers and irregular eaters) and compared the difference between each sex. It is a very interesting issue. However, the way to expression in result part is not easy to read and to fully understand. Even if I read the article and compared the tables for several times, I couldn't remember the conclusion. Moreover, I think the statistic analysis could be improved. 

Author Response

This is a cross-sectional study which is associated with the meal patterns of older adults. Researchers have categorized older adults into three meal patterns types (regular eaters, meal skippers and irregular eaters) and compared the difference between each sex. It is a very interesting issue. However, the way to expression in result part is not easy to read and to fully understand. Even if I read the article and compared the tables for several times, I couldn't remember the conclusion. Moreover, I think the statistic analysis could be improved. 

Response: We have copy-edited the entire manuscript for better readability. We further have completely rewritten the conclusion. We did not quite grasp the comment on the statistical analyses, hence we left it unchanged.

Reviewer 3 Report

The paper of Stalling et al. presents an interesting work about the potential influence of meal patterns in human health.

In general, the study meets the requirements established by Nutrients journal. The title is concise. The abstract summarizes the most important findings in approximately 200 words; the maximum allowed. The Introduction section provides a background which is well-written and defines the objective of the work. The procedures and methodologies performed in the study are well-described and explained. The study sample is representative as it includes an appropriate number of individuals (1198 participants). Results are interesting and could be valuable for future research. All tables are appropriately used and necessary for the correct understanding of the results. The Discussion section provides an adequate interpretation of the most important findings. References are properly present in the text and references section.

In conclusion, the present manuscript is worthy of publication after minor corrections. The specific comments and suggestions for the improvement of the manuscript are included below:

(1) According to the Nutrients rules, a concise cover letter should have been submitted with view to explaining the interest and originality of the study and, above all, the reasons for why the work should be published.

(2)   Conclusions should be enriched. This section must be extended as it partially summarizes the results obtained. Please, provide a more detailed and complete paragraph.

(3) Although the authors have explained the limitations of this study in the Discussion, it must be clearly indicated it in the Conclusions section. Given that these important limitations (e.g., study design; assessment of meal patterns with a self-developed questionnaire not validated), authors must not generalize these results.

Author Response

The paper of Stalling et al. presents an interesting work about the potential influence of meal patterns in human health.

In general, the study meets the requirements established by Nutrients journal. The title is concise. The abstract summarizes the most important findings in approximately 200 words; the maximum allowed. The Introduction section provides a background which is well-written and defines the objective of the work. The procedures and methodologies performed in the study are well-described and explained. The study sample is representative as it includes an appropriate number of individuals (1198 participants). Results are interesting and could be valuable for future research. All tables are appropriately used and necessary for the correct understanding of the results. The Discussion section provides an adequate interpretation of the most important findings. References are properly present in the text and references section.

In conclusion, the present manuscript is worthy of publication after minor corrections. The specific comments and suggestions for the improvement of the manuscript are included below:

(1) According to the Nutrients rules, a concise cover letter should have been submitted with view to explaining the interest and originality of the study and, above all, the reasons for why the work should be published.

Response: We have adapted the cover letter, so it now details the aim and originality of the study.

(2)   Conclusions should be enriched. This section must be extended as it partially summarizes the results obtained. Please, provide a more detailed and complete paragraph.

Response: We have completely rewritten the conclusion for a better summary of the results and therefore a better understanding.

(3) Although the authors have explained the limitations of this study in the Discussion, it must be clearly indicated it in the Conclusions section. Given that these important limitations (e.g., study design; assessment of meal patterns with a self-developed questionnaire not validated), authors must not generalize these results.

Response: Thank you for your comment. We agree and have added the limitations of this study to the conclusions section.

This manuscript is a resubmission of an earlier submission. The following is a list of the peer review reports and author responses from that submission.

Round 1

Reviewer 1 Report

Congratulations on the article you wrote, I think it's a very interesting topic, it's well written and easy to understand. Some suggestions to improve:

In the methodology, you explain the meal patterns and food frequency. In the second paragraph where you talk about FFQ, you write that  “The variables whole-grain products, green vegetables, meat, and alcohol were in- 93 cluded in this study" - question - you included only these food groups? Why? I think that was important to justify why you choose this groups, and did not include fruit or fish for example!

The discussion is ok.

Author Response

Reviewer 1

Congratulations on the article you wrote, I think it's a very interesting topic, it's well written and easy to understand. Some suggestions to improve:

In the methodology, you explain the meal patterns and food frequency. In the second paragraph where you talk about FFQ, you write that “The variables whole-grain products, green vegetables, meat, and alcohol were included in this study" - question - you included only these food groups? Why? I think that was important to justify why you choose this groups, and did not include fruit or fish for example!

The discussion is ok.

Response: Thank you very much for your comment. Since the question on food frequency was originally developed to assess bone health, it comprised the following food items: hard cheese, soft cheese, milk/ buttermilk, and yogurt/ kefir, eggs, whole-grain products, legumes, cabbage/ green vegetables, meat, fat fish, alcohol, and mineral water. To cover carbohydrates, vegetables, protein, fats, the variables whole-grain products, cabbage/ green vegetables, and meat were included in this study. Furthermore, alcohol consumption was included as a potential health compromising behaviour. We have clarified this in the manuscript in the methods section.

Reviewer 2 Report

In the present study, the authors investigated the meal patterns of older adults in Germany and their associated sociodemographic factors and health indicators. Overall, the results are interesting. However, I have some doubts about the credibility and comprehensiveness of this research. As discussed in the text, the cross-sectional study design limits the final conclusions of the article, and longitudinal studies are needed to investigate possible associations between dietary patterns and sociodemographic factors and health indicators. At the same time, it is difficult to define meal skippers and irregular eaters, and the investigators themselves may not know which type of people they belong to. Finally, the relationship between dietary patterns and health in older adults may be more difficult to grasp due to the prevalence of underlying diseases.

Author Response

Reviewer 2

In the present study, the authors investigated the meal patterns of older adults in Germany and their associated sociodemographic factors and health indicators. Overall, the results are interesting. However, I have some doubts about the credibility and comprehensiveness of this research. As discussed in the text, the cross-sectional study design limits the final conclusions of the article, and longitudinal studies are needed to investigate possible associations between dietary patterns and sociodemographic factors and health indicators. At the same time, it is difficult to define meal skippers and irregular eaters, and the investigators themselves may not know which type of people they belong to. Finally, the relationship between dietary patterns and health in older adults may be more difficult to grasp due to the prevalence of underlying diseases.

Response: Thank you very much for your comments. We have adapted the manuscript regarding the definition of meal patterns and the description of the used methods. We hope, that this clarifies some issues.

Reviewer 3 Report

Summary of manuscript: This was a cross-sectional study that aimed to assess meal patterns in older adults. The meal patterns were characterized as follows: regular eaters, meal skippers, and irregular eaters. The results demonstrated that most men and women were regular eaters, followed by meal skippers and irregular eaters. It was concluded that there are possible associations between meal patterns and sociodemographic and health factors. In addition, the results can provide an opportunity for disease prevention and health promotion of older adults.   

General comments: I carefully reviewed this manuscript. The authors provided a study with interesting results. I recommend including more details on the definitions of the meal patterns and the questionnaires. I provided my specific comments below.    

Introduction

Point 1: It was mentioned that “knowledge regarding older adults is still scarce.” Are there other attributes that make your study unique?

Materials and Methods

Point 2: Line 66: Should “individual” be plural?

Point 3: Lines 81-85: If someone consumes at least 3 meals a day, but only on the weekends, are they considered regular eaters? If someone only skips one meal a day, but only on the weekends, are they considered meal skippers? If the answers are yes, then a meal skipper might consume more 3 meals a day during the week than a regular eater. Additional clarifications of the meal patterns are needed in this paragraph. A table might be helpful.

Point 4: Lines 93-94: Please explain why these items were included in the study. What does “meat” include?  

Point 5: Are all of the questionnaires and surveys available for the readers to access, such as the “paper-pencil” questionnaires, meal pattern questionnaires, and the food frequency questionnaires? Are they validated? The readers might want more details of these items. Please mention these topics in the Materials and Methods.

Results

Point 6: Line 161: Should “hat” be had?

Point 7: Table 2: Under the “Men” section, there is a large space between “Mostly or” and “always”. Furthermore, it’s difficult to differentiate between the various items in the left column. Please make it easier to emphasize the different items, such as self-prepared meals, whole-grain products, green vegetables, and so forth. I recommend revising the left columns of the other tables, as well.

Point 8: Table 3: In the “Women” and “Men” sections, there is a large space between “Very good or” and “ex”.

Discussion

Point 9: Lines 195-198: This sentence relates to the questionnaires used in this study. Please provide more details. Did the questionnaires differ between your study and other studies?  

Point 10: Lines 236-238 and Lines 239-240: As mentioned above, please provide more information on these questionnaires. Are they available for the readers to view?

Author Response

Reviewer 3

Summary of manuscript: This was a cross-sectional study that aimed to assess meal patterns in older adults. The meal patterns were characterized as follows: regular eaters, meal skippers, and irregular eaters. The results demonstrated that most men and women were regular eaters, followed by meal skippers and irregular eaters. It was concluded that there are possible associations between meal patterns and sociodemographic and health factors. In addition, the results can provide an opportunity for disease prevention and health promotion of older adults.   

General comments: I carefully reviewed this manuscript. The authors provided a study with interesting results. I recommend including more details on the definitions of the meal patterns and the questionnaires. I provided my specific comments below.    

Response: Thank you very much for all your comments and suggestions. We have adapted the manuscript accordingly and hope that it has improved.

Introduction

Point 1: It was mentioned that “knowledge regarding older adults is still scarce.” Are there other attributes that make your study unique?

Response: We find that, because of limited research on this topic, having more knowledge on meal patterns of older adults the most important aim of this study, since it can be beneficial for preventing a wide range of non-communicable diseases and health risks, and can further help in developing health promotion interventions.

Materials and Methods

Point 2: Line 66: Should “individual” be plural?

Response: Yes, thank you for noticing this error. We have corrected this and it now says “individuals”.

Point 3: Lines 81-85: If someone consumes at least 3 meals a day, but only on the weekends, are they considered regular eaters? If someone only skips one meal a day, but only on the weekends, are they considered meal skippers? If the answers are yes, then a meal skipper might consume more 3 meals a day during the week than a regular eater. Additional clarifications of the meal patterns are needed in this paragraph. A table might be helpful.

Response: Yes, technically participants who consume at least 3 meals a day but only on weekends would be considered regular eaters, since we recoded “only on weekends” and “only on weekdays” to “several times a week”. We did this, because the two options were only checked by few participants and only one person checked “only on weekends” for all three main meals. However, this participant additionally answered, that precise answers were difficult to give, because of eating irregularly. Therefore, she was categorised as a meal skipper.

Point 4: Lines 93-94: Please explain why these items were included in the study. What does “meat” include?  

Response: Since the question on food frequency was originally developed to assess bone health, it comprised the following food items: hard cheese, soft cheese, milk/ buttermilk, and yogurt/ kefir, eggs, whole-grain products, legumes, cabbage/ green vegetables, meat, fat fish, alcohol, and mineral water. To cover carbohydrates, vegetables, protein, fats, the variables whole-grain products, cabbage/ green vegetables, and meat were included in this study. Meat was not further specified. Furthermore, alcohol consumption was included as a potential health compromising behaviour. We have clarified this in the manuscript in the methods section.

Point 5: Are all of the questionnaires and surveys available for the readers to access, such as the “paper-pencil” questionnaires, meal pattern questionnaires, and the food frequency questionnaires? Are they validated? The readers might want more details of these items. Please mention these topics in the Materials and Methods.

Response: Only one questionnaire was used (self-administered paper-pencil questionnaire), that included question for assessing meal patterns and food consumption frequency. Some of the used questions are validated and some are self-developed, with no validation study. We have clarified this in the manuscript and have added, that the questionnaire and survey materials are available on request from the corresponding author. 

Results

Point 6: Line 161: Should “hat” be had?

Response: Yes, we have corrected this mistake.

Point 7: Table 2: Under the “Men” section, there is a large space between “Mostly or” and “always”. Furthermore, it’s difficult to differentiate between the various items in the left column. Please make it easier to emphasize the different items, such as self-prepared meals, whole-grain products, green vegetables, and so forth. I recommend revising the left columns of the other tables, as well.

Response: When transferring the manuscript to the template, some formatting issues occurred, which is why it was difficult to differentiate in the tables. We have formatted all tables accordingly, so it should be easier now.

Point 8: Table 3: In the “Women” and “Men” sections, there is a large space between “Very good or” and “ex”.

Response: This was also due to the formatting issues described in the answer above. We have formatted the table as well.

Discussion

Point 9: Lines 195-198: This sentence relates to the questionnaires used in this study. Please provide more details. Did the questionnaires differ between your study and other studies?  

Response: Yes, they differed. For example, one study used a diary-based assessment, whereas we assessed meal pattern via one question using a self-administered questionnaire. We have added this fact and more details to the manuscript.

Point 10: Lines 236-238 and Lines 239-240: As mentioned above, please provide more information on these questionnaires. Are they available for the readers to view?

Response: They are available on request. For further information, please see response to point 5.

Round 2

Reviewer 3 Report

I would like to thank the authors for revising the manuscript. Unfortunately, I still have concerns regarding the meal pattern definitions. For example, there appears to be overlap between the regular eaters and meal skippers. The regular eaters consumed at least three meals a day at least several times a week. The meal skippers skipped one meal per day at least several times per week. However, both the regular eaters and meal skippers might consume the same number of three meals a day during the week. Additionally, they both might skip the same number of meals during the week. Therefore, the definitions are not clear.